# Comparative In Vitro Activity of Ceftolozane/Tazobactam against Clinical Isolates of *Pseudomonas aeruginosa* and *Enterobacterales* from Five Latin American Countries

**DOI:** 10.3390/antibiotics11081101

**Published:** 2022-08-13

**Authors:** Juan Carlos García-Betancur, Elsa De La Cadena, María F. Mojica, Cristhian Hernández-Gómez, Adriana Correa, Marcela A. Radice, Paulo Castañeda-Méndez, Diego A. Jaime-Villalon, Ana C. Gales, José M. Munita, María Virginia Villegas

**Affiliations:** 1Grupo de Investigación en Resistencia Antimicrobiana y Epidemiología Hospitalaria, Universidad El Bosque, Bogotá 110121, Colombia; 2Department of Molecular Biology and Microbiology, School of Medicine, Case Western Reserve University, Cleveland, OH 44106-7164, USA; 3Veterans Affairs Medical Center for Antimicrobial Resistance and Epidemiology (Case VA CARES), Case Western Reserve University-Cleveland, Cleveland, OH 44106-7164, USA; 4Research Service, VA Northeast Ohio Healthcare System, Cleveland, OH 44106-7164, USA; 5Clínica Imbanaco, Cali 760031, Colombia; 6Facultad de Ciencias Básicas, Universidad Santiago de Cali, Cali 760033, Colombia; 7Laboratorio de Resistencia Bacteriana, Facultad de Farmacia y Bioquímica, IBaViM, Universidad de Buenos Aires, Buenos Aires 70512, Argentina; 8National Scientific and Technical Research Council (CONICET), Godoy Cruz, Buenos Aires 2290, Argentina; 9Hospital Médica Sur, Ciudad de México 14050, Mexico; 10Department of Internal Medicine, Division of Infectious Diseases, Universidade Federal de São Paulo, São Paulo 04039-032, Brazil; 11Millennium Initiative for Collaborative Research on Bacterial Resistance (MICROB-R), Santiago 7650568, Chile; 12Genomics and Resistant Microbes Group, Facultad de Medicina Clínica Alemana, Universidad del Desarrollo, Santiago 8320000, Chile

**Keywords:** ceftolozane/tazobactam, *Pseudomonas aeruginosa*, *Enterobacterales*, antimicrobial resistance, Latin America

## Abstract

Background: Ceftolozane/tazobactam (C/T) is a combination of an antipseudomonal oxyiminoaminothiazolyl cephalosporin with potent in vitro activity against *Pseudomonas aeruginosa* and tazobactam, a known β-lactamase inhibitor. The aim of this study was to evaluate the activity of C/T against clinical isolates of *P. aeruginosa* and *Enterobacterales* collected from five Latin American countries between 2016 and 2017, before its clinical use in Latin America, and to compare it with the activity of other available broad-spectrum antimicrobial agents. Methods: a total of 2760 clinical isolates (508 *P. aeruginosa* and 2252 *Enterobacterales*) were consecutively collected from 20 hospitals and susceptibility to C/T and comparator agents was tested and interpreted following the current guidelines. Results: according to the CLSI breakpoints, 68.1% (346/508) of *P. aeruginosa* and 83.9% (1889/2252) of *Enterobacterales* isolates were susceptible to C/T. Overall, C/T demonstrated higher in vitro activity than currently available cephalosporins, piperacillin/tazobactam and carbapenems when tested against *P. aeruginosa*, and its performance in vitro was comparable to fosfomycin. When tested against *Enterobacterales*, it showed higher activity than cephalosporins and piperacillin/tazobactam, and similar activity to ertapenem. Conclusions: these results show that C/T is an active β-lactam agent against clinical isolates of *P. aeruginosa* and *Enterobacterales*.

## 1. Introduction

Antibiotic resistance is a global public health problem that compromises the effectiveness of antimicrobial therapy in healthcare settings and in the community, threatening the enormous gains made by the discovery of new antibiotics [1,2]. This scenario is particularly relevant for Gram-negative pathogens such as *P. aeruginosa* and *Enterobacterales.* These bacteria are a frequent cause of healthcare-associated infections (HAI) and are among the most common pathogenic organisms acquiring resistance to extended-spectrum cephalosporins and carbapenems, which are the preferred antimicrobial regimens [3,4]. Indeed, once they develop resistance to broad-spectrum β-lactam antibiotics, reliable therapeutic options become scarce.

Ceftolozane/tazobactam (C/T) is a combination of a last-generation cephalosporin, ceftolozane, and a well-known β-lactamase inhibitor, tazobactam [5]. The antibacterial mechanism of ceftolozane, as for any other β-lactam, is based on the inhibition of the bacterial cell wall biosynthesis, specifically mediated by its binding to penicillin-binding proteins (PBPs). Ceftolozane inhibits all of the essential PBPs produced by *P. aeruginosa*: PBP1b, PBP1c, PBP3 and some of *Enterobacterales*, like the PBP3 from *Escherichia coli* [6]. Tazobactam, although having clinically irrelevant in vitro activity against bacterial PBPs, is a potent and irreversible inhibitor of some β-lactamases, which restores the antibacterial activity of ceftolozane and improves the spectrum of activity against ESBL-producing *Enterobacterales* and some anaerobes [7]. Structurally, the substitution of a pyrazole side chain on the 3-position of the cephem ring improves the outer membrane permeability of ceftolozane and grants increased stability against some AmpC β-lactamases, leading to improved activity against P. aeruginosa [8]. In 2014, the U.S. Food and Drug Administration (FDA) approved this combination for the treatment of complicated urinary tract infections, and also, in combination with metronidazole, for the treatment of complicated intra-abdominal infections. In 2019, C/T was also approved for the treatment of hospital-acquired (HAP) and ventilator-acquired (VAP) pneumonia [9]. Ceftolozane has been reported to have improved outer membrane permeability, is less affected by the activity of efflux pumps, and has enhanced stability against the chromosomal *Pseudomonas*-derived cephalosporinase (PDC), resulting in potent in vitro activity against *P. aeruginosa* [10,11]. Hence, C/T remains highly active against most Gram-negative bacteria (GNB) including multidrug-resistant (MDR) *P. aeruginosa* (i.e., non-susceptible to ≥1 antimicrobial agent in ≥3 different antimicrobial categories) [12], but it has lower in vitro activity than carbapenems against ESBL-producing *K. pneumoniae* and, in addition, it is not active against carbapenemase-producing isolates independent of the carbapenemase class produced [13,14,15,16].

By documenting local, regional, and global epidemiological patterns and trends of antimicrobial resistance, surveillance programs play a fundamental role in the design of strategies to combat the dissemination and management of MDR pathogens. Studies describing the in vitro activity of C/T and comparator agents against clinical isolates of *Enterobacterales* and *P. aeruginosa* have shown that the susceptibility to C/T varies greatly among different regions. For example, while susceptibility to C/T among isolates collected from different European hospitals between 2011 and 2012 was 84.5% for *P. aeruginosa* and 91.5% for *Enterobacterales*, in Canada, the susceptibility to C/T in *P. aeruginosa* in 2013 was >98%, similar to the U.S. and some Western European countries, where susceptibility to this antibiotic was >94% for *P. aeruginosa* and 90% for *Enterobacterales* [17,18,19]. In the Latin American region, the susceptibility of clinical isolates collected between 2013 and 2015 showed that 89.6% of *P. aeruginosa* and 85.9% of *Enterobacterales* were susceptible to C/T [20]. Due to the increasing global spread of MDR GNB, up-to-date surveillance studies are necessary to closely monitor this impact on the use of new antibiotics such as C/T.

Herein, we extend the information from previous studies, report the activity of C/T against clinical isolates of *P. aeruginosa* and *Enterobacterales* retrieved from 20 different Latin American hospitals from January 2016 through October 2017, and compare this to the activity of several broad-spectrum antimicrobials. Our results provide new data on the susceptibility patterns of C/T in this geographical area, which are characterized as being endemic for several antimicrobial-resistance mechanisms [21].

## 2. Results

### 2.1. Susceptibility Profile for Phenotypic Subsets

To facilitate analysis of the data, we grouped the *P. aeruginosa* and *Enterobacterales* isolates in phenotypic groups. As shown in Table 1, for *P. aeruginosa*, the piperacillin/tazobactam-NS phenotypic subset was observed in 45.5% (231/508), ceftazidime-NS in 46.8% (238/508), and meropenem-NS in 53.5% (272/508) of the isolates.

For *Enterobacterales,* in *E. coli,* the ESBL non-CRE phenotype was observed in 30.2% (425/1409) of the isolates and in 25.1% (153/610) of *K. pneumoniae*. The ertapenem-susceptible phenotype was observed in 70.3% (64/91) of *S. marcescens*, 63.4% (71/112) of *E. cloacae* complex, and 73.3% (22/30) of *K. aerogenes* isolates, as shown in Table 1. In addition, AmpC derepression, as indicated by resistance to at least one of the third-generation cephalosporins tested (cefotaxime, ceftriaxone, or ceftazidime) and ertapenem susceptibility, was observed in 31.9% (29/91) of *S. marcescens*, 28.6% (32/112) of *E. cloacae* complex, and 53.3% (16/30) of *K. aerogenes* isolates.

### 2.2. Activity of C/T and Comparator Agents against P. aeruginosa

According to the current CLSI M100 guideline breakpoints, 31.9% (162/508) of the *P. aeruginosa* isolates were non-susceptible to C/T. Relevant phenotypic subsets and their distribution among the five Latin American countries are shown in Table 2. In general, *P. aeruginosa* exhibited a moderate susceptibility to C/T (68.1%; MIC_50_ 2 mg/L, MIC_90_ >128 mg/L). C/T MIC values for *P. aeruginosa* ranged from 1 mg/L to >128 mg/L, as shown in Figure 1. C/T demonstrated better in vitro activity than any other evaluated β-lactam, including carbapenems, and was similar to fosfomycin (susceptibility: 72.2%; MIC_50_, 64 mg/L and MIC_90_, >128 mg/L; Table 1).

Susceptibility rates to C/T in the piperacillin/tazobactam-NS subset and in the ceftazidime-NS subset were particularly low (36.8% and 35.7%, respectively). Similarly, susceptibility to carbapenems was particularly low in these isolates, not exceeding 11% for meropenem in the ceftazidime-NS subset (Table 1). For these phenotypic subsets, fosfomycin showed a moderate susceptibility, ranging from 40.8% in the ceftazidime-NS subset to 62.8% in the piperacillin/tazobactam-NS subset.

When *P. aeruginosa* clinical isolates were analyzed by country, differences in the susceptibility rates to C/T and comparator agents arose, as seen in Figure 1 and Table 2. The highest susceptibility rates were found in Chile (80.6%; MIC_50_ 2 mg/L, MIC_90_ 32 mg/L), followed by Argentina (70%; MIC_50_ 4 mg/L, MIC_90_ > 128 mg/L), Brazil (68.3%; MIC_50_ 2 mg/L, MIC_90_ > 128 mg/L), Colombia (66.1%; MIC_50_ < 1 mg/L, MIC_90_ > 128 mg/L) and finally, Mexico (64.4%; MIC_50_ 2 mg/L, MIC_90_ > 128 mg/L).

### 2.3. Activity of C/T and Comparator Agents against Enterobacterales

The distribution of the 2252 clinical isolates of *Enterobacterales* within each relevant phenotypic subset, and their susceptibility to C/T and comparator antimicrobials, are shown in Figure 2 and Table 1. Overall, C/T displayed a good activity against *Enterobacterales*, with activity against 73.8% (1662/2252) of the isolates. As shown in Figure 2, C/T MIC values ranged from <1 mg/L to >128 mg/L, with MIC_50_ and MIC_90_ being 1 mg/L and 32 mg/L, respectively. The highest susceptibility rates to C/T among *Enterobacterales* were found in *E. coli* (93.2%; MIC_50_ 1 mg/L, MIC_90_ 1 mg/L), followed by *K. aerogenes* (73.3%; MIC_50_ 1 mg/L, MIC_90_ 16 mg/L), *S. marcescens* (71.4%; MIC_50_ 1 mg/L, MIC_90_ 64 mg/L), *K. pneumoniae* (68.7%; MIC_50_ 1 mg/L, MIC_90_ 128 mg/L) and finally, the *E. cloacae* complex (62.5%; MIC_50_ 1 mg/L, MIC_90_ 32 mg/L).

In comparison to other antimicrobial agents, C/T demonstrated higher activity than currently available cephalosporins (susceptibility: 58.8% for ceftazidime, 47.7% for ceftriaxone and 47.6% for cefotaxime), and even piperacillin/tazobactam (susceptibility 70.8%). On the other hand, C/T activity was similar to ertapenem’s (susceptibility 73.5%), but inferior to other carbapenems (susceptibility to doripenem 82.8%, to meropenem 82.6% and imipenem 78.7%). In contrast, tigecycline (87.9%) and fosfomycin (91.6%) susceptibility was higher than C/T (73.8%).

When data were stratified by species, doripenem showed the highest susceptibility rates against *E. coli* (95.7%), while fosfomycin was the most active in vitro antimicrobial agent against *K. pneumoniae*, *S. marcescens* and *K. aerogenes* (susceptibility: 92.5%, 94.5% and 96.7%, respectively). Tigecycline was the most active antimicrobial drug against *E. cloacae* complex (susceptibility 89.3%), as shown in Table 1**.**

As shown in Appendix A, susceptibility rates of *Enterobacterales* to C/T and comparator antimicrobial agents varied considerably among the five Latin American countries included in this study. In general terms, susceptibility to C/T reached 61.3% in Brazil, 64.3% in Mexico, 71.6% in Colombia, 85.6% in Chile and 88.6% in Argentina. MIC frequency distribution of C/T for all *Enterobacterales*, in every participating country, is shown in Figure 1.

Evaluating each species individually, C/T showed the best activity rates against *E. coli*, with susceptibility ranging between 80% (Brazil) and 95.6% (Argentina). For *K. pneumoniae*, this antimicrobial showed a lower activity, not surpassing a susceptibility rate of 80% (in Mexico). For both species, C/T performed better than ceftazidime and piperacillin/tazobactam in all five countries. Despite the variability among countries, C/T performed similarly to carbapenems, tigecycline and fosfomycin in *E. coli*. On the other hand, carbapenems, tigecycline and fosfomycin displayed better activity than C/T against *K. pneumoniae.* For *E. cloacae* complex, *K. aerogenes* and *S.*
*marcescens* susceptibility rates to C/T and comparator agents varied greatly among the five evaluated countries. Detailed information for each pathogen, their susceptibility rates, relevant phenotypic subsets and their distribution among the five countries is displayed in Appendix A.

## 3. Discussion

The potent combination of C/T was conceived to circumvent the therapeutic challenges imposed by the concomitant presence of the *Pseudomonas*-derived cephalosporinase (PDC), ESBLs and intrinsic efflux pump systems [5,16,23,24]. Moreover, C/T also has potent activity against *Enterobacterales* strains, carrying widely spread ESBLs and AmpC β-lactamases, such as CTX-M-15 and CMY-2, respectively. Therefore, C/T constitutes a carbapenem-sparing option for the clinical treatment of infections caused by these common pathogens. Importantly, C/T lacks activity against all carbapenemases [16,25,26]. Unfortunately, resistance to C/T dramatically narrows treatment options for *P. aeruginosa*, and despite possible susceptibility to fosfomycin, this antimicrobial has several clinical restrictions, and are limited to combination therapy regimens [27,28].

The results presented here are important for Latin America given the extended dissemination of antimicrobial resistance in this geographical area [21,29,30,31], the endemicity of carbapenem-resistant bacteria in some of these countries [21,32,33], and the limited availability of effective antibiotic therapies for MDR *P. aeruginosa* and *Enterobacterales* [34,35]. Epidemiological studies surveying the resistant status of C/T against a set of other antimicrobial agents for nosocomial pathogens retrieve important information that allows for the determination of country-specific and even institution-specific activity of antimicrobial drugs, allowing clinicians to improve empiric therapy, as well as allowing for the design of effective antimicrobial stewardship strategies.

Compared with similar studies carried out in previous years [20], we found a decrease in the susceptibility of C/T in *P. aeruginosa*. Indeed, during the period between 2013 and 2015, susceptibility of this pathogen to C/T was reported in 82.4%, while our results show a susceptibility decrease to 68.1%. Pfaller et al. surveyed 12 hospitals (vs. 20 in our study) in 4 countries (not including Colombia), reporting a higher susceptibility rate to C/T. The lower susceptibilities to C/T found in Colombia might contribute to this result. Several publications of *P. aeruginosa* strains in Colombia reported the production of carbapenemases, such as KPC and VIM, and even co-resistance of KPC/VIM [21,32,33]. This difference, added to the dissemination of other carbapenem resistance determinants among *P. aeruginosa* and *Enterobacterales* strains in Latin America, might have contributed to the decreased C/T susceptibility reported in this study. Our data also showed an important reduction in C/T susceptibility when compared to the global study reported from 2015 to 2017 by Shortridge et al. [13], where the susceptibility to C/T in the Latin American region (represented by isolates from Colombia and Mexico) was found to be 90.8%. Furthermore, several studies have reported a significant increase in the VIM carbapenemase in Mexico, as recently reviewed [21]. Regional differences in the presence of β-lactam resistance determinants, including ESBL and carbapenemases [13,16,17,20,26,28,36,37,38,39,40,41], may explain the differences between studies. Therefore, the understanding of the molecular mechanisms involved in the resistant phenotypes and the decrease in susceptibility to C/T in Latin America, particularly for *P. aeruginosa*, is fundamental to improve therapeutic and stewardship strategies in this geographical region.

Despite the lower susceptibility reported in our study, C/T is still the most active β-lactam antibiotic against *P. aeruginosa*. Likewise, our study confirms that C/T is highly active against *Enterobacterales*, with similar susceptibility rates to carbapenems for *E. coli* and *S. marcescens*, and slightly lower rates for *K. pneumoniae, E. cloacae* complex, and *K. aerogenes*. Our data support the idea that C/T could provide a reasonable therapeutic option for Latin American hospitals with high rates of MDR *P. aeruginosa* and ESBL-producing *Enterobacterales* [41].

In particular, it is important to highlight that the susceptibility to C/T in β-lactam resistant *P. aeruginosa*, such as meropenem-NS, was particularly low, with a mean value of 46.7%, and varying among countries (Chile 60.9% and Colombia 38.9%). These resistant phenotypes represented 53.5% of all *P. aeruginosa* isolates in this Latin American survey. Molecular analyses are underway to determine the mechanisms of resistance, such as mutations leading to structural modifications and/or overexpression of AmpC, OprD loss, PDC upregulation, the presence of carbapenemases, or amino acid substitutions, insertions or deletions found in PDC variants [24,26,42,43]. These results also highlight the imperative need for routine susceptibility testing prior to its use.

The results from this study also show the importance of frequent regional epidemiological surveys for new therapeutic options, due to the dynamic resistance of *P. aeruginosa* and *Enterobacterales*.

## 4. Materials and Methods

### 4.1. Sampling Sites and Organisms

A total of 2760 GNB clinical isolates, including *Escherichia coli* (*n* = 1409, 51%), *Klebsiella pneumoniae ss. pneumoniae* (*n* = 610, 22%), *Enterobacter cloacae* (*n* = 112, 4%), *Serratia marcescens* (*n* = 91, 3%), *Klebsiella aerogenes* (*n* = 30, 1%) and *P. aeruginosa* (*n* = 508, 18%), were collected in 5 Latin American countries (Argentina, Brazil, Chile, Colombia and Mexico) at 20 medical facilities during the study period, from January 2016 to October 2017, prior to the approval of C/T in this region. Species identification was performed locally at each participating medical center and reidentified at Clínica Imbanaco (Cali, Colombia) using MALDI-TOF MS (BioMérieux, Marcy-l’Étoile, France). The list of the principal species tested against C/T by country, is presented in Appendix A.

### 4.2. Antimicrobial Susceptibility Testing

Determination of minimum inhibitory concentrations (MIC) was performed using broth microdilution with customized *Sensititre* plates (TREK Diagnostic Systems, Westlake, OH, USA), and following the CLSI M100 guidelines 2020 [44], where C/T breakpoints are ≤2 mg/L for *Enterobacterales* and ≤4 mg/L for *P. aeruginosa*. These breakpoints were determined based on a C/T dosage of 3 g q 8 h for pneumonia, and of 1.5 g q 8 h for any other indication for both *Enterobacterales* and *P. aeruginosa*, according to the CLSI guidelines [44]. C/T concentrations are expressed as absolute numbers representing only the concentration of ceftolozane, given that tazobactam remains constant in 4 mg/L. Quality control (QC) was performed in accordance to the CLSI M07-A10 and M100-S27 documents, using the following strains: *E. coli* ATCC 25922 and *P. aeruginosa* ATCC 27853. Except for fosfomycin and tigecycline, results were interpreted according to the CLSI M100 breakpoints [44]. Fosfomycin susceptibility results were interpreted according to available breakpoints set by CLSI (susceptible at an MIC of ≤64 mg/L for *E. coli* and extrapolated to other *Enterobacterales*) and EUCAST for *P. aeruginosa* (ECOFF of ≤128 mg/L) [22]. The U.S. FDA product package insert criteria were used as the breakpoint for tigecycline [22,44]. Colistin was not considered as a comparator agent in this study, since there are many conflicting results; *Sensititre* plates have shown a high rate of false-resistant results [45] and this method is not recommended by the CLSI or EUCAST for antimicrobial susceptibility testing with this antibiotic.

### 4.3. Phenotypic Subsets

Isolates belonging to the order of *Enterobacterales* were classified in phenotypic subsets based on their MIC profile, as follows: ESBL non-CRE phenotype was defined as *E. coli* or *K. pneumoniae* isolates, displaying a MIC ≥ 2 mg/L for ceftriaxone and MIC < 1 mg/L for ertapenem. In addition, AmpC producers (*Serratia marcescens*, *Enterobacter cloacae* complex, and *Klebsiella aerogenes*) were classified as ertapenem-susceptible (MIC < 1 mg/L) and cefotaxime-resistant (MIC ≥ 4 mg/L). *P. aeruginosa* isolates were classified as piperacillin/tazobactam-non-susceptible (NS) (MIC ≥ 32 mg/L), meropenem-susceptible (MIC ≤ 2 mg/L), meropenem-NS (MIC ≥ 4 mg/L) and ceftazidime-NS (MIC ≥ 16 mg/L).

## Figures and Tables

**Figure 1 antibiotics-11-01101-f001:**
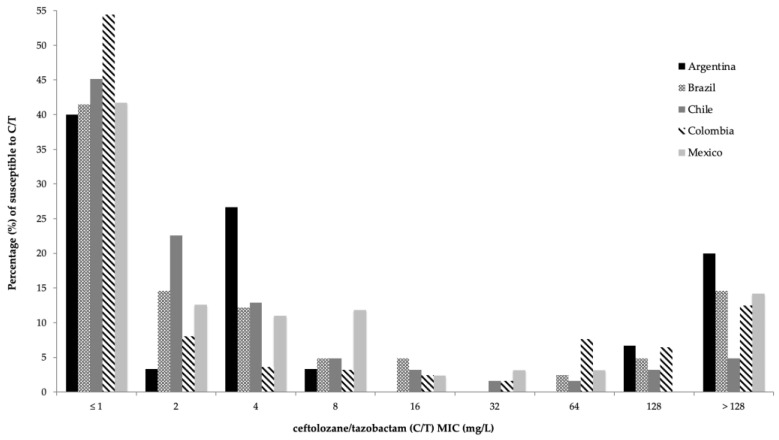
Susceptibility to C/T of clinical isolates of *P. aeruginosa* in five Latin American countries between 2016 and 2017 in relation to the minimum inhibitory concentration (MIC) to C/T. Each bar represents one of the five countries evaluated.

**Figure 2 antibiotics-11-01101-f002:**
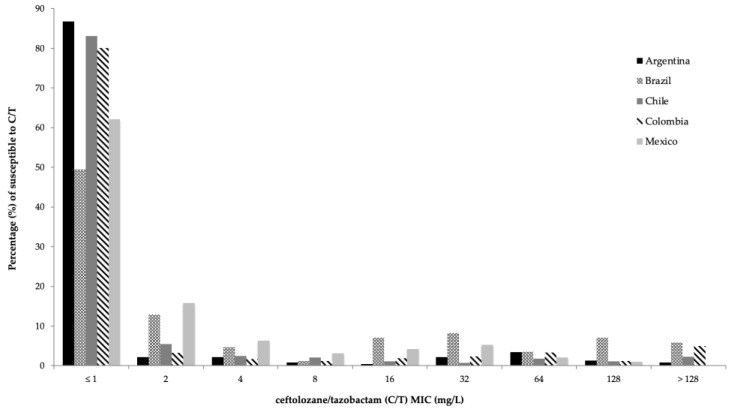
Susceptibility to C/T of clinical isolates of *Enterobacterales* in five Latin American countries between 2016 and 2017 in relation to the minimum inhibitory concentration (MIC) to C/T. Each bar represents one of the five countries evaluated.

**Table 1 antibiotics-11-01101-t001:** Susceptibility of phenotypic subsets of clinical isolates of *P. aeruginosa* and *Enterobacterales* to C/T and comparator agents between 2016 and 2017 from five Latin American countries.

		Percentage of Susceptibility (%)										
Organism	Number of Isolates	C/T	CRO	CTX	CAZ	TZP	ETP	IMI	MEM	DOR	TGC	FOS
*Pseudomonas aeruginosa*	508	68.1	NA	NA	53.1	54.5	NA	10.8	36.4	38.6	NA	72.2 **
piperacillin/tazobactam-NS	231	36.8	NA	NA	10.9	-	NA	3.46	10.8	13.4	NA	62.8 **
ceftazidime-NS	238	35.7	NA	NA	-	13.4	NA	14.7	21.0	26.1	NA	40.8 **
meropenem-NS	272	46.7	NA	NA	30.9	29.8	NA	1.1	-	12.5	NA	64.3 **
*Escherichia coli*	1409	93.2	60.3	60.9	71.8	90.3	90.3	94.3	95.5	95.7	95.6	94.8
ESBL non-CRE phenotype *	425	95.3	-	3.5	33.4	90.3	-	99.5	100.0	100.0	98.1	90.8
*Klebsiella pneumoniae*	610	68.7	44.9	46.9	50.0	59.2	70.0	71.0	76.6	77.7	84.4	92.5
ESBL non-CRE phenotype *	153	79.7	-	8.5	24.2	52.3	-	97.4	99.3	100.0	96.1	94.8
*Serratia marcescens*	91	71.4	46.2	42.9	62.6	69.2	70.3	73.6	78.0	76.9	80.2	94.5
ertapenem-susceptible	64	92.2	64.0	59.4	85.9	90.6	-	93.7	100.0	98.4	90.6	100.0
*Enterobacter cloacae* complex	112	62.5	30.4	30.4	42.9	51.8	63.4	71.4	79.5	80.4	89.3	79.5
ertapenem-susceptible	71	84.5	45.1	45.1	63.4	79.4	-	90.1	100.0	98.59	94.4	83.1
*Klebsiella aerogenes*	30	73.3	56.7	56.7	66.7	83.3	73.3	83.3	83.3	83.33	90.0	96.6
ertapenem-susceptible	22	90.9	77.3	72.7	86.4	95.4	-	100.0	100.0	100.0	100.0	100.0

C/T: ceftolozane/tazobactam, CRO: ceftriaxone, CTX: cefotaxime, CAZ: ceftazidime, TZP: piperacillin/tazobactam, ETP: ertapenem, IMI: imipenem, MEM: meropenem, DOR: doripenem, TGC: tigecycline, FOS: fosfomycin. * Ceftriaxone-non-susceptible (MIC >2 mg/L) and ertapenem-susceptible (MIC <1 mg/L) were used as ESBL phenotype indicator. ** ECOFF ≤128 mg/L [22].

**Table 2 antibiotics-11-01101-t002:** Susceptibility of *P. aeruginosa* clinical isolates and phenotypic subsets to C/T and comparator agents between 2016 and 2017 from five Latin American countries.

			Percentage of Susceptibility (%)						
	Organism	Number of Isolates	C/T	CAZ	TZP	IMI	MEM	DOR	FOS *
	*Pseudomonas aeruginosa*	30	70.0	33.3	40.0	30.0	40.0	53.3	53.3
	piperacillin/tazobactam-NS	18	50.0	5.6	-	5.6	5.6	22.2	38.9
Argentina	meropenem-susceptible	12	66.7	91.7	91.7	75.0	100.0	100.0	75.0
	meropenem-NS	18	50.0	11.1	5.6	0.0	-	22.2	38.9
	ceftazidime-NS	20	55.0	-	15.0	20.0	15.0	35.0	45.0
	*Pseudomonas aeruginosa*	41	68.3	49.6	46.3	12.2	26.8	29.3	73.2
	piperacillin/tazobactam-NS	22	50.0	40.9	-	13.6	13.6	9.1	40.9
Brazil	meropenem-susceptible	11	100.0	90.9	72.7	36.4	100.0	72.7	72.7
	meropenem-NS	30	56.7	53.3	36.7	3.3	-	13.3	73.3
	ceftazidime-NS	15	33.3	-	13.3	6.7	6.7	6.7	66.7
	*Pseudomonas aeruginosa*	62	80.6	56.5	58.1	41.9	62.9	66.1	83.9
	piperacillin/tazobactam-NS	26	61.5	11.5	-	26.9	30.8	34.6	80.8
Chile	meropenem-susceptible	39	92.3	76.9	79.5	64.1	100.0	97.4	84.6
	meropenem-NS	23	60.9	21.7	21.7	4.3	-	13.0	82.6
	ceftazidime-NS	27	55.6	-	14.8	25.9	33.3	37.0	85.2
	*Pseudomonas aeruginosa*	248	66.1	54.8	58.9	30.2	47.2	52.4	80.2
	piperacillin/tazobactam-NS	102	23.5	7.8	-	4.9	9.8	15.7	76.5
Colombia	meropenem-susceptible	117	96.6	97.5	103.3	71.5	100.0	70.8	65.1
	meropenem-NS	131	38.9	26.7	29.8	0.8	-	13.7	73.3
	ceftazidime-NS	112	26.8	-	16.1	11.6	14.3	21.4	75.9
	*Pseudomonas aeruginosa*	127	64.4	49.6	50.4	25.2	44.9	42.5	55.1
	piperacillin/tazobactam-NS	63	39.7	6.3	-	14.3	28.6	27.0	39.7
Mexico	meropenem-susceptible	57	84.2	64.9	45.6	56.1	100.0	86.0	68.4
	meropenem-NS	70	50.0	37.1	35.7	0.0	-	7.1	44.3
	ceftazidime-NS	64	37.5	-	3.1	15.6	31.3	31.3	40.6

C/T: ceftolozane/tazobactam, CAZ: ceftazidime, TZP: piperacillin/tazobactam, IMI: imipenem, MEM: meropenem, DOR: doripenem, FOS: fosfomycin. * ECOFF ≤ 128 mg/L [22].

## Data Availability

The data presented in this study are available on request from the corresponding author. It is not publicly available due to the presence of unpublished data on the original databases.

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
