# Peer review of "Comparative In Vitro Activity of Ceftolozane/Tazobactam against Clinical Isolates of Pseudomonas aeruginosa and Enterobacterales from Five Latin American Countries"

_antibiotics, 2022, doi:10.3390/antibiotics11081101_

Round 1

Reviewer 1 Report

In this paper, the in vitro efficacy of ceftolozane/tazobactam against P. aeruginosa and Enterobacterales clinical isolates from South America is described. This study is an important report of the efficacy of these antibiotics in that particular part of the globe. It is not clear why, in figure 1 and figure 2, the authors chose to represent MICs as fractions (eg. 1/4 mg/L, 16/4 mg/L etc.) and not as simple numbers. Looking at paragraph 2.2 it also looks like these values are not fractions, so they should be interpreted as 1 mg/mL, 16 mg/mL etc. But then the fractions are back in paragraph 2.3. This can be very confusing, it would be best if the authors reported their results in a uniform fashion.

Author Response

In this paper, the in vitro efficacy of ceftolozane/tazobactam against P. aeruginosa and Enterobacterales clinical isolates from South America is described. This study is an important report of the efficacy of these antibiotics in that particular part of the globe.

1. It is not clear why, in figure 1 and figure 2, the authors chose to represent MICs as fractions (eg. 1/4 mg/L, 16/4 mg/L etc.) and not as simple numbers. Looking at paragraph 2.2 it also looks like these values are not fractions, so they should be interpreted as 1 mg/mL, 16 mg/mL etc. But then the fractions are back in paragraph 2.3. This can be very confusing, it would be best if the authors reported their results in a uniform fashion.

For Figure 1 and Figure 2, as well as in paragraphs 2.2 and 2.3, we present the concentrations of ceftolozane and tazobactam as a fraction, indicating the concentration of the cephalosporin ceftolozane in the numerator and the concentration of the β-lactamase inhibitor tazobactam in the denominator. As the reviewer noticed, in some parts of the text we mentioned C/T concentrations as a fraction and, in some parts as an absolute number. We have corrected this issue. In agreement with the reviewer, and also taking into consideration that the concentration of tazobactam remains constant (4 mg/L) for all ceftolozane concentrations, we replaced all fractions for absolute numbers, representing only the concentration (in mg/L) of ceftolozane. Replacements can be observed in the revised version of the manuscript in Figure 1 (lines 197 to 201), Figure 2 (lines 202 to 206), as well as in paragraph 2.2 (lines 116 to 117) and in paragraph 2.3 (lines 142 to 145). Additionally, in paragraph 4.2 from the section Materials and Methods, we have also corrected this annotation and, we have cleared that C/T concentrations will be expressed as absolute numbers representing only the concentration of ceftolozane, given that tazobactam remains constant in 4 mg/L (lines 290 to 293).

Reviewer 2 Report

In the study: “Comparative in vitro activity of ceftolozane/tazobactam against clinical isolates of Pseudomonas aeruginosa and Enterobacter- ales from five Latin American countries”. The results and information mentioned are interesting, and the data are to certain extent sufficient to support the conclusion. However, there are still some issues need to be addressed.

1. In the “Antimicrobial susceptibility testing” part, the specific dosage of Ceftolozane/tazobactam (C/T) should be provided.

2. About the antimicrobial test. Only the determination of minimum inhibitory concentrations (MIC) were carried out in the manuscript, which are really insufficient. It would be better to perform more tests such as the antimicrobial activity assay (e.g. Carbon. 2018, 130, 775-781), the morphological characterization of bacteria (e.g. e.g. Appl. Catal. B. 2022, 301, 120826; Carbon. 2019, 155, 397-402).

3. What is the antibacterial mechanism of C/T? Does it have long-term and stable antibacterial properties?

Author Response

In the study: “Comparative in vitro activity of ceftolozane/tazobactam against clinical isolates of Pseudomonas aeruginosa and Enterobacter- ales from five Latin American countries”. The results and information mentioned are interesting, and the data are to certain extent sufficient to support the conclusion. However, there are still some issues need to be addressed.

1.In the “Antimicrobial susceptibility testing” part, the specific dosage of Ceftolozane/tazobactam (C/T) should be provided.

Following the recommendation made by the reviewer, in lines 289 to 291 we have added the dosage of C/T for which the breakpoints for Enterobacterales and P. aeruginosa were determined. This dosage corresponds to:

3 g q 8 h for pneumonia.

1.5 g q 8 h for any other indication.

These breakpoints and dosage for both, Enterobacterales and P. aeruginosa, are referenced in: P. C. and L. S. I. Wayne, Performance Standards for Antimicrobial Testing. CLSI supplement M100., 30th ed. Wayne, PA., 2022. [1] This reference was added to the manuscript in line 291.

  1. About the antimicrobial test. Only the determination of minimum inhibitory concentrations (MIC) were carried out in the manuscript, which are really insufficient. It would be better to perform more tests such as the antimicrobial activity assay (e.g. Carbon. 2018, 130, 775-781), the morphological characterization of bacteria (e.g. e.g. Appl. Catal. B. 2022, 301, 120826; Carbon. 2019, 155, 397-402).

Although, methods to assay antibacterial activities of chemical compounds, like the paper-disk diffusion method (DDM) used by Xia et al. (Carbon, 130, 2018, 775-781) or the plate count method (or agar dilution method) used by Meng et al. (Applied Catalysis B: Environmental, February 2022, 120826) might be recommended for some specific antibacterial molecules under some specific conditions (for example, DDM for fosfomycin); the diameter of the inhibition zone obtained by DDM do not offer a quantitative MIC value. It only offers a measure of i) Resistant ii) Intermediate or iii) Susceptible. In addition, it should be mentioned that DDM is not an appropriate method for all kind of molecules, since the solubility factor of the antimicrobial significantly affects the diffusion from a filter paper disk into the agar medium, altering the results and its interpretation. On the other hand, the agar dilution method is a highly laborious method not suitable for the antimicrobial susceptibility testing of 2760 clinical isolates. Finally, and more importantly, given its utility in the clinical laboratory, its replicability and scalability, the quantitative determination of MIC by broth microdilution (BMD), is the reference method for antimicrobial susceptibility testing for these in vitro studies [2]. In addition, the Clinical and Laboratory Standard Institute (CLSI) and European Committee on Antimicrobial Susceptibility Testing (EUCAST) recommend broth microdilution (BMD) as the reference method for MIC determination for most antibiotics, C/T included [1], [3].

  1. What is the antibacterial mechanism of C/T? Does it have long-term and stable antibacterial properties?

As cited in lines 55 to 56, and expanded in lines 61 to 65: “Ceftolozane/tazobactam (C/T) is a combination of a cephalosporin of last generation, ceftolozane, with a well-known β-lactamase inhibitor, tazobactam [5].…Ceftolozane has been reported to have higher affinity for penicillin binding proteins PBP1b, PBP1c, and PBP3, improved outer membrane permeability, is less affected by the activity of efflux pumps, and has enhanced stability against the chromosomal Pseudomonas derived cephalosporinase (PDC), resulting in potent in vitro activity against P. aeruginosa [7], [8].”

Nevertheless, and intended to follow the reviewer’s recommendations, we have rephrased the introduction in lines 55 to 67, adding some ideas as follows: “The antibacterial mechanism of ceftolozane, as for any other β-lactam, is based on the inhibition of the bacterial cell wall biosynthesis, specifically mediated by its binding to penicillin-binding proteins (PBPs). Ceftolozane inhibits all of the essential PBPs produced by P. aeruginosa: PBP1b, PBP1c, PBP3 and those of Enterobacterales, like the PBP3 from Escherichia coli [4]. Tazobactam, although with clinically irrelevant in vitro activity against bacterial PBPs, it is a potent and irreversible inhibitor of some β -lactamases, which restores the antibacterial activity of ceftolozane and improves the spectrum of activity against ESBL-producing Enterobacterales and some anaerobes [5]. Structurally, the substitution of a pyrazole side chain on the 3-position of the cephem ring, improves the outer membrane permeability of ceftolozane and grants increased stability against some AmpC β -lactamases, leading to improved activity against P. aeruginosa [6].

 References:

[1]             P. C. and L. S. I. Wayne, Performance Standards for Antimicrobial Testing. CLSI supplement M100., 32nd ed. 2022.

[2]             D. R. Giacobbe et al., “Ceftolozane/tazobactam: place in therapy,” Expert Rev. Anti. Infect. Ther., vol. 16, no. 4, pp. 307–320, 2018, doi: 10.1080/14787210.2018.1447381.

[3]             EUCAST, “The European Committee on Antimicrobial Susceptibility Testing. Breakpoint tables for interpretation of MICs and zone diameters.,” http://www.eucast.org, pp. 1–110, 2022.

[4]             B. Moya, L. Zamorano, C. Juan, J. L. Pérez, Y. Ge, and A. Oliver, “Activity of a new cephalosporin, CXA-101 (FR264205), against β-lactam-resistant Pseudomonas aeruginosa mutants selected in vitro and after antipseudomonal treatment of intensive care unit patients,” Antimicrob. Agents Chemother., vol. 54, no. 3, pp. 1213–1217, 2010, doi: 10.1128/AAC.01104-09.

[5]             K. Murano et al., “Structural requirements for the stability of novel cephalosporins to AmpC β-lactamase based on 3D-structure,” Bioorganic Med. Chem., vol. 16, no. 5, pp. 2261–2275, 2008, doi: 10.1016/j.bmc.2007.11.074.

[6]             A. Toda et al., “Synthesis and SAR of novel parenteral anti-pseudomonal cephalosporins: Discovery of FR264205,” Bioorganic Med. Chem. Lett., vol. 18, no. 17, pp. 4849–4852, 2008, doi: 10.1016/j.bmcl.2008.07.085.
